# *BRAF* Mutation in Colorectal Cancers: From Prognostic Marker to Targetable Mutation

**DOI:** 10.3390/cancers12113236

**Published:** 2020-11-03

**Authors:** Izuma Nakayama, Toru Hirota, Eiji Shinozaki

**Affiliations:** 1Department of Gastroenterological Chemotherapy, Cancer Institute Hospital of the Japanese Foundation for Cancer Research (JFCR), Tokyo 135-8550, Japan; 2Department of Experimental Pathology, Cancer Institute of the Japanese Foundation for Cancer Research (JFCR), Tokyo 135-8550, Japan; thirota@jfcr.or.jp

**Keywords:** RAF–MEK–ERK signaling pathway, *BRAF* inhibitor, cancer precision medicine

## Abstract

**Simple Summary:**

Colorectal cancer with a mutation in an oncogene BRAF has paid much attention, as it comprises a population with dismal prognosis since two decades ago. A series of research since then has successfully changed this malignancy to be treatable with specific treatment. Here we thoroughly overviewed the basic, translational and clinical studies on colorectal cancer with BRAF mutation from a physician’s viewpoint. Accumulating lines of evidence suggest that intervention of the trunk cellular growth signal transduction pathway, namely EGFR-RAS-RAF-MEK-ERK pathway, is a clue to controlling this disease. However, it is not so straightforward. Recent studies unveil the diverse and plastic nature of this signal transduction pathway. We will introduce our endeavor to conquer this condition, based on newly arriving datasets, and discuss how we could open the door to future development of CRC treatment.

**Abstract:**

The Raf murine sarcoma viral oncogene homolog B (*BRAF*) mutation is detected in 8–12% of metastatic colorectal cancers (mCRCs) and is strongly correlated with poor prognosis. The recent success of the BEACON CRC study and the development of targeted therapy have led to the determination of *BRAF*-mutated mCRCs as an independent category. For nearly two decades, a growing body of evidence has established the significance of the *BRAF* mutation in the development of CRC. Herein, we overview both basic and clinical data relevant to *BRAF*-mutated CRC, mainly focusing on the development of treatment strategies. This review is organized into eight sections, including clinicopathological features, molecular features, prognosis, the predictive value of anti-epidermal growth factor receptor (EGFR) therapy, resistant mechanisms for *BRAF*-targeting treatment, the heterogeneity of the *BRAF* mutation, future perspectives, and conclusions. A characterization of the canonical mitogen-activated protein kinase (MAPK) pathway is essential for controlling this malignancy, and the optimal combination of multiple interventions for treatments remains a point of debate.

## 1. Introduction

Raf murine sarcoma viral oncogene homolog B (*BRAF*)-mutated colorectal cancers (CRCs) are found in a subgroup of CRCs with distinctive clinicopathological features [1]. Within a decade, emerging evidence on the *BRAF* mutation led clinicians to recognize the prognostic and predictive value of this gene alteration. Recently, the *BRAF* mutation has come into consideration when deliberating possible treatment options in clinical practice [2,3]. Currently, targeted therapy for *BRAF*-mutated metastatic CRC (mCRC) has been established [4,5]. In this review, we summarize the findings that have shaped our current understanding of the *BRAF* mutation.

Progress in general basic oncology has accelerated the transition of the significance of the *BRAF* mutation into clinical practice. We also introduce recent developments in cancer precision medicine, which would serve as a tailwind for the widespread adoption of *BRAF* genetic testing [6,7]. Moreover, we refer to the key findings from basic science which rationally support the foundation of current clinical trials [8,9].

Finally, we mention the future perspectives of *BRAF*-mutated mCRC and extend our discussion beyond the *BRAF* gene, toward the comprehensive assessment of the RAS–RAF–MEK–MAPK pathway.

*BRAF*-mutated CRC is currently receiving attention not only from clinicians, but also basic scientists, and thus numerous reviews have been published [3,8,9,10,11,12,13,14,15,16]. As a medical oncology team, we would like to highlight *BRAF*-mutated CRC from the clinician’s viewpoint.

## 2. Dawn of Targeted Therapy for *BRAF*-Mutated CRC

Two articles on the current *RAF* kinase genes were first published in 1983 [8,17,18], marking the beginning of such research. At this time, there were three RAF proteins—ARAF, BRAF, and CRAF—in mammalian cells known to display serine/threonine kinase activity [8,18]. Within the first decade of the discovery of *RAF* kinase, several studies have identified the function of RAF family proteins and their association with cancer. The RAF family proteins were shown to be activated by GTP-bound RAS and work as the effector to activate the signal transduction of the RAS–RAF–MEK–MAPK pathway, leading to cellular proliferation, differentiation, migration, and survival [8,9]. The RAS–RAF–MEK–MAPK pathway is dysregulated in many cancers. The constitutive activation of this signaling pathway occurs in oncogenic RAS- and RAF-driven cancers. In 2002, Davis et al. reported a high frequency of the *BRAF* mutation in human cancers, including melanoma, lung, and colorectal cancers [19]. Their findings highlighted *BRAF*-mutated CRC and marked the dawn of exploring targeted therapy for *BRAF*-mutated CRC.

## 3. Distinctive Characteristics of *BRAF*-Mutated CRC: Molecular and Clinicopathological Aspects

The *BRAF* mutation is detected in 8–12% of mCRCs and the *BRAF* gene encodes 766 amino acids [3]. The most prevalent point mutation occurs in the activation A-loop, near V600, and *BRAF*^V600E^ accounts for >90% of mutations [3,8]. The positive association with microsatellite instability-high (MSI-H)/deficient mismatch repair (dMMR) tumors and mutual exclusiveness with the *KRAS* mutation of *BRAF^V600E^*-mutated CRC were initially well documented [20]. The *BRAF*^V600E^ mutation was found in MSI-H/dMMR tumors with higher incidences, which were reported to range from 8% to 78% [20,21,22,23,24,25,26,27,28]. Hypermethylation is also one of the molecular features of *BRAF*^V600E^-mutated CRC. The *BRAF*^V600E^ mutation was frequently observed more frequently in CpG island methylation phenotype (CIMP)-high CRC (77%), when compared to CIMP-low (18%) or negative (0%) CRC, with statistically significant differences [21]. These overlapping molecular features of *BRAF*^V600E^ between MSI and CIMP would be organized by the understanding of the molecular pathogenesis of the serrated pathway [29]. Several studies recurrently identified the *BRAF*^V600E^ mutation in precursor lesions, such as traditional serrated adenoma [21,29,30]. The *BRAF* mutation is thought to be the earliest event occurring in a precancerous lesion in the serrated pathway. Subsequently, the methylation of the CpG island at the promoter lesion would lead to the silencing of tumor suppressor genes, resulting in carcinogenesis [29]. Therefore, *BRAF*^V600E^-mutated CRC with sporadic microsatellite instability occurs as a consequence of the methylation of MutL homolog1 (MLH1) [29,30,31]. This is why a higher incidence of CpG island methylation is commonly seen in *BRAF*^V600E^-mutated CRC, regardless of the MSI status [22].

Several studies have consistently reported that *BRAF*^V600E^-mutated CRC had distinctive clinicopathological features. *BRAF*^V600E^-mutated CRCs were observed to be more prevalent in elderly or female patients and right-sided primary or mucinous histology tumors [1,2,3,4,5,14,18,19,20,21,22,23,24,25,26,27,28,29,30,31,32]. In addition, several studies reported that *BRAF*-mutated CRC patients were more frequently observed in Caucasian than Asian or African American individuals [33,34,35,36]. Indeed, the prevalence of *BRAF*-mutated CRC was relatively low (6.4%) in the Japanese Nationwide Cancer Genome Screening Project (SCRUM-Japan), compared to those of large-scale studies mainly conducted in Western countries (8–12%) [3,37]. It is known that there are distinctive features in the metastatic site between *BRAF*^V600E^-mutated CRC and others. In general, the liver is the most prevalent metastatic site of CRC, but *BRAF*^V600E^-mutated CRCs tend to metastasize to the peritoneum, rather than the liver or lung [24,38]. However, these clinicopathological features are commonly seen in MSI-H or CIMP-high tumors [39,40]. MSI-H or CIMP-high tumors and *BRAF*^V600E^ shared these features which cannot be attributed to their subtype.

In summary, clinicians at that time were aware of *BRAF*-mutated CRCs as a distinctive subtype of CRCs. Accumulating evidence in the initial period of research had significance for a profound understanding of the etiology, which is different from traditional adenoma–carcinoma sequencing [41]. At that time, the choice of treatment for CRC made by clinicians was not affected by *BRAF* status. The *BRAF* mutation is more frequently observed in sporadic CRC with a hypermethylated phenotype, but not in hereditary CRC, such as the Lynch syndrome. The clinical utility of *BRAF* genetic testing had only been found in the relatively convenient discriminator between sporadic and hereditary CRC [42,43].

## 4. *BRAF* Mutation Recognized as a Negative Prognostic Marker

We now discuss the prognostic impact of *BRAF* mutation in CRC patients. As is often the case, the crucial factor for the genetic testing of *BRAF* in clinical practice depends on whether the presence of the *BRAF* mutation affects clinicians’ decision making. First, we provide an overview of the association between the mutation and the indication of adjuvant chemotherapy for patients who underwent surgery with a curative intent. A retrospective cohort study conducted in multiple facilities in the Netherlands demonstrated that the *BRAF* mutation is an independent prognostic factor for overall survival (OS) (hazard ratio (HR) 2.22, 95% confidence interval (CI) 1.25–4.00), disease-free survival (DFS) (HR 2.33, 95% CI 1.22–4.55), and cancer-specific survival (CCS) (HR 2.13, 95% CI 1.01–4.55) in Stage Ⅱ/Ⅲ CRC by multivariate analysis [44]. However, as this study included patients treated between 1996 and 2004 and postoperative chemotherapy was not mentioned, this result cannot be directly applied to current practices. Two major clinical trials conducted in Europe and the United States reported consistent results: the retrospective study of the PETACC-3, EORTC 40993, SAKK 60-00 trials showed on one hand an independent negative prognostic value of the *BRAF* mutation in Stage Ⅱ/Ⅲ CRC (HR 1.78, 95% CI 1.15–2.76) [40]. On the other hand, the negative impact of the *BRAF* mutation on recurrence after curative resection was not statistically significant (HR 1.30, 95% CI 0.87–1.95) [45]. The other study using the results from the CALGB 89803 trial also provided an inferior effect on the survival of Stage Ⅲ CRC patients (HR 1.66, 95% CI 1.05–2.63) [46]. Thus, the negative prognostic impact of *BRAF* mutation on survival is reproducible, but the *BRAF* mutation is not useful as a negative predictive marker for recurrence after curative resection. It should be noted that these clinical trials assessed the efficacy of adding irinotecan to the fluoropyrimidine in the adjuvant setting, while oxaliplatin-based adjuvant chemotherapy had later been established as a standard regimen for Stage Ⅲ CRC patients who had received curative resection [47,48,49,50,51]. Therefore, the point becomes whether the *BRAF* mutation is indicative in making decisions on the adjuvant oxaliplatin-containing chemotherapy. The molecular profile of patients treated in another three pivotal clinical trials (MOSAIC, NSABP-C 07, and -C 08) was assessed and the significance of the *BRAF* mutation for survival and recurrence was evaluated [52,53]. However, their results were inconsistent: in a pooled analysis of NSABP-C 07 and -C 08, the *BRAF* mutation had a significant association with poor OS (HR 1.46, 95% CI 1.20–1.79) and survival after relapse (HR 2.31, 95% CI 1.83–2.95) [52], but in the MOSAIC study, a poor prognostic value of the *BRAF* mutation was not demonstrated. In clinical practice, however, oxaliplatin in addition to fluorouracil plus leucovorin were used, even in *BRAF*-mutated CRCs [53]. Thereby, oxaliplatin-containing adjuvant chemotherapy has been recommended for Stage Ⅲ CRC patients after curative resection, irrespectively of *BRAF* status, and to date, *BRAF* mutation has failed to become a game changer in the treatment of CRCs after curative resection.

Then, we will discuss the prognostic value of the *BRAF* mutation after metastasectomy. The resection of metastatic sites in general had been demonstrated to be key in the long-term survival or cure of mCRC patients, especially in cases of colorectal cancer liver-limited metastasis (CRLM) [54]. That said, tumor relapse is to occur in 50–75% of patients after R0 metastasectomy [55,56]. Therefore, a biomarker indicating the risk of recurrence is warranted. The prognostic value of the *KRAS* mutation in patients with resectable mCRCs has been pointed out [57,58,59,60,61,62,63]. That of *BRAF* mutation has not yet reached conclusive results to date, primarily because a very limited number of *BRAF*-mutated CRCs are subject to resection, unlike in cases of *KRAS*-mutated CRC (30–40%) [64]. The overall incidence of the *BRAF* mutation in mCRC is reported to be 8–12% [1] and recent studies show that patients with the *BRAF* mutation account for only 1–4% of mCRC patients who underwent metastasectomy [38,59,64,65,66,67,68,69]. Due to these very small numbers, the impact of the *BRAF* mutation on survival with other confounders remains unclear [59,70]. For instance, there are only 24 mCRC patients harboring the *BRAF* mutation in a systematic review and meta-analysis [68]. All reports on metastasectomy which we reviewed focused on hepatectomy, except for two articles [38,71]: Schweiger et al. could not find any *BRAF* mutation in mCRC patients who underwent lung metastasectomy (*n* = 44) [72]. The large population on lung metastasectomy for *BRAF*-mutated mCRC (*n* = 19) was reported by Renaud et al. [71]. There, the median overall survival time of patients with a *BRAF^V600E^* mutation was estimated to be 15.0 months (95% CI 12.2–17.8) and the survival rate at 5 years was 0% [71]. Yaeger reported that four patients underwent resection of lung metastases and all had a relapse within 20 months [38]. Thus, none of the reported *BRAF*-mutated mCRC patients that underwent lung metastasectomy (*n* = 23) were cured [38,71]. Along with this, these meta-analyses of hepatectomy showed the negative prognostic impact of *BRAF* mutation on survival, with statistical significance HR 3.06 and 95% CI 1.79–5.20 [68], and HR 3.90 and 95% CI 1.96–7.73 [73]. It is worth noting that these studies included the *BRAF V600E* and *BRAF non-V600E* mutations. A recently published study revealed the different prognostic impact between the *BRAF V600E* and *non-V600E* mutation in CRLM patients who had undergone hepatectomy. Patients without the *BRAF*^non-V600E^ mutation, but with the *BRAF*^V600E^ mutation, had an association with worse OS (HR 2.76, 95% CI 1.74–4.37, *p* < 0.001) and DFS (HR 2.04, 95% CI 1.30–3.20, *p* = 0.002) when compared to the patients with *BRAF* wild type [64]. As mentioned above, *BRAF* mutation seems to be associated with distinctive clinicopathological features, such as older age, females, mucinous histology, or right-sided primary tumors. While some of these factors may contribute to survival outcome, due to the small sample size, multivariate analysis could not sufficiently eliminate these confounders. Case-matched analyses were conducted to compensate previous studies, and consistently demonstrated the negative prognostic significance, but outcomes after the relapse varied [64,68]. In contrast, Bachet et al. demonstrated that the *BRAF* mutation does not have an association with worse DFS [74]. They reported a shorter survival time after relapse in *BRAF*-mutated mCRC (23.0 months, 95% CI 11.0–35.0) than in the wild type (44.3 months, 95% CI 35.9–52.6) [74]. Metastasectomy maintained a positive impact on survival, both in the presence and absence of the *BRAF* mutation [65]. The examination of the BRAF status could be recognized as one reliable prognostic value but has not been accepted as a predictive marker in the selection for metastasectomy.

In metastatic disease, the *BRAF* mutation has been widely accepted to have a strong negative impact on survival [2,3,22,75,76,77,78,79,80]. For those without K-ras mutation, treatment with an anti-EGFR monoclonal antibody (mAb) is the preferred choice; however, the status of *BRAF* should be considered when evaluating the prognostic value, because *BRAF* is located downstream of EGFR in the signal transduction pathway. In addition to being a prognostic factor, *BRAF* mutation will also be a predictive marker for anti-EGFR mAb therapy, which will be further discussed in the next section. In a recently conducted comprehensive analysis by Seigmann and colleagues, they focused on the *BRAF*-mutated mCRC patients treated using chemotherapy without anti-EGFR mAb. From datasets with a large sample size, they demonstrated the shorter OS of *BRAF*-mutated mCRC patients (10.8 vs. 16.4 months, HR 1.49 (95% CI 1.23–1.80), *p* < 0.001). Their result was consistent with the data from the other RCTs in the patients who did not undergo anti-EGFR mAb treatment [78,79,81]. Seligmann et al. showed similar disease control rates and progression free survival (PFS) between *BRAF* mutant-type and wild-type mCRCs in a first-line treatment setting, however, shorter PFS in the subsequent line and the lower rate of patients received the later-line treatment [2,82]. Thus, the inferior OS of *BRAF*-mutated mCRC is largely attributable to post-progression survival. [2,82]. However, the shorter PFSs of *BRAF*-mutated mCRC in first-line treatment were reported in some studies [81,83]. Despite these results being inconsistent with those of the efficacy of the standard doublet first-line regimen, a shorter OS of *BRAF*-mutated mCRC was consistently reported.

Additionally, the prognostic impact of the *BRAF* mutation was assessed in the context of its correlation with *KRAS*, MSI status, and CIMP, which made it difficult to interpret the prognostic impact of the *BRAF* mutation. An informative study by Phipps and colleagues [84] addressed the associations of the *BRAF* mutation with *KRAS*, MSI, and CIMP in a well-organized manner, in mainly resectable-stage CRCs. Their results indicated three key points: (a) MSI-H revealed a better survival impact, even for the *BRAF*-mutated CRC; (b) both *KRAS-* and *BRAF*-mutated CRC had a poor survival impact, but the prognostic impact of *KRAS* was weaker than that of *BRAF*; (c) CIMP status could be representative of the *BRAF* mutation or MSI-H in consideration of the prognostic impact. Therefore, MSI-H comprises a good prognosis group (even better without *BRAF* mutation), but the non-MSS group increasingly worsened with the *KRAS* mutation and further, with the *BRAF* mutation. As mentioned above, *BRAF*-mutated CRCs often overlap with MSI-H [20,21,22,23,24,25,26,27,28]; therefore, it is important to discuss the association of the *BRAF* mutation with MSI status in the prognosis of the *BRAF*-mutated CRC patients [85,86]. In MSS CRCs, a large-scale pooled analysis of prospective randomized trials (*n* = 3278) pointed out that *BRAF*-mutated CRC revealed a significantly worse association with OS (HR 1.84, 95% CI 1.14–2.97), but not with RFS (HR 1.36, 95% CI 0.86–2.16) in Stage Ⅱ/Ⅲ disease [45]. Likewise, in the meta-analysis of four clinical trials (*n* = 3603), statistically significantly worse outcomes were detected when focusing on the proficient mismatch repair (pMMR) (OS: HR 1.94, 95% CI 1.57–2.40, PFS: HR 1.34, 95% CI 1.10–1.64) [25]. Therefore, it seems reasonable to conclude that the *BRAF* mutation indicates poor prognosis in MSS/pMMR CRC [3,25,28]. In MSI-H CRCs, controversial conclusions were drawn: some studies concluded that the *BRAF* mutation did not have an association with a poor prognosis in MSI-H/dMMR group CRC [22,25,26,27,28,45,87,88], but the others concluded that the negative impact of the *BRAF* mutation was observed regardless of MSI/MMR status [24,46,89]. At first glance, these conclusions seem controversial, but the data themselves may not be very different. Due to the small number of *BRAF*-mutated CRC with MSI-H/dMMR and a favorable prognosis of MSI-H/dMMR, statistical power is to be limited, and these studies might have underestimated the negative prognosis of *BRAF*-mutated CRCs with MSI-H/dMMR. Although many studies analyzed all stages of CRCs together [22,26,27,87], an inverse effect of the MSI status on survival between the early and metastatic stages should be considered. Namely, MSI-H/dMMR CRCs are positive prognostic markers in Stage Ⅱ/Ⅲ, and negative in the metastatic stage [28]. Thus, the prognostic value of the *BRAF* mutation with MSI-H/dMMR should be dealt separately between resectable and metastatic stages, in theory; however, in reality, this will divide a small population into small pieces. These are the current technical limitations associated with the prognostic value of *BRAF*-mutated mCRC with MSI-H /dMMR.

In summary, although clinicians appear to share the idea that the *BRAF* mutation is associated with poor prognosis, the prognostic impact of the *BRAF* mutation alone cannot be the only consideration in decision making in clinical practice. Indeed, mandatory *BRAF* genetic testing was not recommended in the therapeutic guideline released in 2012 [90]. However, the dismal outcomes of *BRAF*-mutated mCRC patients were always a motivation of those developing therapeutics. As described in following section, the 2016 ESMO consensus guideline recommends the assessment of the *BRAF* mutation status along with the emergence of therapeutic strategies for BRAF-mutated mCRC patients [91].

## 5. *BRAF* Status Required as a Predictive Marker in Clinical Decision Making

Cetuximab (Cmab) and Panitumumab (Pmab) have been shown to have excellent effectiveness for the treatment of chemotherapy-resistant mCRC [92,93,94]. After these two anti-EGFR mAbs emerged in the clinical practice of mCRC, a new era of targeted therapy for the EGF/EGFR signaling pathway began. In 2006, Lievre et al. reported that the *KRAS* mutation potentially had a predictive value in the treatment of anti-EGFR mAb [95]. A significant role of the *KRAS* mutation as a negative predictive marker for anti-EGFR mAb had been proven by using tumor samples collected in randomized phase Ⅲ clinical trials. The *KRAS* mutation had been established as a negative predictor for the anti-EGFR antibody [96,97] and the *KRAS* mutation had become the first molecular marker for patient selection in the treatment of mCRC. While the *KRAS* genetic test had a high specificity and nearly 95% of patients harboring the *KRAS* mutation had no benefit from anti-EGFR mAb, it had insufficient efficacy. Approximately 40–60% of patients with *KRAS* wild type could not receive a clinical response to anti-EGFR therapy [98].

*BRAF* plays a key role as the effector molecule of *KRAS* in the activation of the *RAS/RAF* signaling pathway. Therefore, researchers focused on the *BRAF* mutation to elucidate its predictive value for the treatment of anti-EGFR therapy. Retrospective single-arm studies primarily showed that the *BRAF* mutation might be a negative predictive marker for the efficacy of Cmab or Pmab. No responder of anti-EGFR therapy was found in patients with the *BRAF^V600E^* mutation and shorter PFS and OS were observed in the *BRAF*-mutated patients compared to wild-type patients. These three analyses included a very limited number of *BRAF*-mutated CRC patients (*n* = 11, *n* = 5, and *n* = 11, respectively) [99,100,101]. The results were reproduced in a larger size cohort. In the *KRAS* wild-type cohort, *BRAF*-mutated mCRC patients who received C/Pmab monotherapy or C/Pmab plus chemotherapy had a significantly worse prognosis compared to those without the mutation (overall response rate (ORR): 8.3% vs. 38.0%, median PFS: 8 vs. 26 weeks, HR 3.74, 95% CI 2.44–5.75, *p* < 0.0001, median OS: 26 vs. 54 weeks, HR 3.03, 95% CI 1.98–4.63, *p* < 0.0001) [102]. Explorations of the predictive value of the *BRAF* mutation were conducted using the clinical specimens collected in randomized control trials (RCTs) [103,104,105,106,107,108,109,110,111,112]. Two meta-analyses of RCTs had already been published [108,109] and there were other meta-analyses that contained not only RCTs, but also retrospective studies, in the unselected population [113,114,115,116,117,118]. The former two meta-analyses used almost the same RCT data. Rowland et al. excluded two RCTs—NORDIC and FIRE-3—and included the data of *BRAF* wild-type patients to evaluate the efficacy of adding anti-EGFR mAb to backbone chemotherapy or the best supportive care [113]. The statistically significant benefits of adding anti-EGFR mAb in OS and PFS were consistently not proven in both Pietrantonio’s study (OS; HR 0.91, 95% CI 0.62–1.34) and Rowland’s study [113,119]. However, they drew the different conclusions. Pietrantonio et al. simply supported to avoid the administration of anti-EGFR for *BRAF*-mutated mCRC patients [119]. Rowland et al. pointed out not there were insufficient data to discard anti-EGFR mAbs from *BRAF*-mutated mCRC patients [113]. The relatively favorable trends of adding anti-EGFR mAbs were seen in a first-line setting compared to the later line. Both OS and PFS HRs were derived from the three RCTs, PRIME, and a pooled analysis of CRYSTAL and OPUS, were 0.90 (95% CI 0.46–1.16) and 0.58 (95% CI 0.29–1.15) (PRIME) and 0.62 (95% CI 0.36–1.06) and 0.67 (95% CI 0.34–1.29) (CRYSTAL and OPUS) [107,113,114]. The mutation analysis was prospectively conducted in the PRIME study and these RCTs were well designed and regarded as reliable data. On the other hand, the detrimental effect of adding Pmab to irinotecan for the treatment of *KRAS* wild/*BRAF* mutant mCRC patients in a second line setting was reported [110]. Based on these results, guideline members of CRC in the European Society of Medical Oncology (ESMO) committee concluded there was insufficient evidence to exclude anti-EGFR therapy for patients with *BRAF* mutant disease [92]. However, a meta-analysis of the effect of anti-EGFR therapy in a first-line setting demonstrated that a significantly decreased response rate was observed in *BRAF^V600E^*-mutated mCRC compared to *BRAF* wild-type [120]. Van Brummelen et al. reviewed eight meta-analyses, including RCTs and studies conducted in an unselected population [112,113,114,115,116,117,118,119,120,121]. They showed that the clinical benefit from anti-EGFR therapy for BRAF-mutated mCRC patients could not be found in terms of OS, PFS, and ORR [121]. Moreover, seven of eight meta-analyses other than Rowland’s demonstrated the statistically significant inferiority of anti-EGFR therapy in *BRAF*-mutated mCRC compared to *BRAF* wild type.

In the same period, the clinical utility of the *BRAF* mutation status came to be recognized in terms of another clinical aspect. The TRIBE study demonstrated the efficacy and safety of the FOLFOXIRI + Bevacizumab (BV) regimen and this triplet regimen came to be one of the standard treatments in mCRC [122,123]. Due to severe toxicity, the triplet regimen was considered to be the optimal treatment choice for select patients with aggressive tumors who expected to fail to receive second-line therapy after disease progression. *BRAF*-mCRC would be recognized as a tumor which had these characteristics. Phase Ⅱ study data of FOLFOXIRI + BV showed encouraging results in terms of the clinical outcome (RR: 72%, median PFS: 11.8 months, and median OS: 24.1 months) for patients with the *BRAF* mutation [124]. Although significant differences in OS, PFS, and ORR could not be observed between patients in the FOLFOXIRI + BV and FOLFIRI + BV group of the TRIBE phase Ⅲ study [125], the median survival time of the experimental group was reported to be 19.0 months, which was numerically longer than for *BRAF*-mutated mCRC patients treated in previous clinical trials (10–15 months) [24,75,79]. Notably, a recent analysis on the prognosis of *BRAF*-mutated mCRC demonstrated a significantly lower rate (33% vs. *BRAF* wild-type 51%, *p* < 0.001) for patients who received second-line chemotherapy after disease progression [2]. As mentioned above, a shorter PFS in the first-line group and post-progression survival (PPS) after the first-line treatment were also rationale for using all the key drugs, including 5-FU, oxaliplatin, and irinotecan, in the initial treatment [2,78]. That is why treatment guidelines encouraged such an up-front aggressive treatment to be considered for *BRAF*-mutated mCRC in clinical practice worldwide [92,126,127].

The cancer precision medicine era became a tailwind in raising awareness of the significance of *BRAF*-mutated CRC in clinical practice. A low incidence of the *BRAF* mutation had been the critical hazard preventing *BRAF* gene testing from entering clinical practice. Next generation sequencing (NGS) technology opened the door for comprehensive molecular analyses of cancer genomes with much higher speed and lower costs [128]. A gene mutation that recurrently appears in cancer genomes can be considered as a “driver mutation” candidate and is, therefore, a target in the precision medicine era. Clinical sequencing in the search for the driver genetic alterations has been widely conducted [6,7,129], and molecular targeting agents, mainly inhibitors, have been developed to suppress these. To overcome the low incidence, umbrella- or basket-type clinical trials enable a patient with a rare driver mutation in a specific organ to efficiently receive the targeted therapy [129], including mCRC patients with the *BRAF* mutation. *BRAF* gene testing had reached the bedside from the bench-side in the wind of cancer precision medicine.

In summary, although the *BRAF* mutation was shown to have a potential negative predictive value for treatment with anti-EGFR antibody, evidence was insufficient to make *BRAF* gene testing mandatory like *RAS* [130,131]. In the same period, several studies demonstrated that patient selection according to *KRAS*, *NRAS*, *BRAF*, and *PIC3CA* mutation could enrich the efficacy of molecular targeted therapy [102,132,133]. The *BRAF* mutation was one of these negative predictors, along with the EGFR pathway. After the success of the TRIBE trial, the triplet regimen emerged as one of the treatment options for aggressive types of CRCs, including *BRAF*-mutated ones. Thereby, the *BRAF* mutation is considered to be an actionable genetic alteration in clinical practice. However, *BRAF*-mutated CRCs had some associations with unfavorable clinical factors for the toxic regimen, such as elderly patients or peritoneum metastasis. Therefore, *BRAF*-mutated patients who actually received the triplet regimen were limited to a highly select population. Specific therapy targeting the *BRAF* mutation was urgently awaited.

## 6. *BRAF* Became a Targetable Molecule in mCRC Treatment

The great success of *BRAF*^V600E^ targeting therapy in melanoma repositioned the *BRAF*^V600E^ mutation from being a prognostic marker to targetable genetic alteration. In retrospect, the history of *BRAF*^V600E^ targeting therapy had indeed begun in 2000, before the first reports on the *BRAF* mutation in human cancer were launched. Sorafenib entered clinical trials as the first RAF inhibitor aimed to treat *RAS*-mutated cancers. However, Sorafenib was a less potent *BRAF*^V600E^ inhibitor because it preferentially bound to the inactive enzyme conformation [8]. The second generation of ATP-competitive RAF inhibitors, e.g., PLX4032 (vemurafenib), which selectively bound to the active form, were developed in 2008, which opened the door to the development of *BRAF*^V600E^ targeting therapy [134]. In 2010, the result of a phase Ⅰ study on vemurafenib were published for the treatment of *BRAF*^V600E^-mutated metastatic melanoma [135]. A phase Ⅱ study demonstrated promising results, and the ORR reached 53% [136]. In 2011, vemurafenib was shown to have improved ORR and PFS compared with a standard treatment (dacarbazine) in a phase Ⅲ RCT, leading to the approval of the US Food and Drug Administration (FDA) [137]. Through these trials for melanoma, *BRAF*^V600E^-targeting agents became available in clinical practice.

Dissimilar to melanoma, however, the efficacy of vemurafenib was not reproduced in the treatment of *BRAF*^V600E^-mutated mCRC. A phase II study showed disappointing results, with an ORR of 5% (95% CI < 1–26), median PFS of 2.1 months (95% CI 1.8–3.6 months), and median OS of 7.7 months (95% CI 3.4–11.6 months) [138]. Observations in melanoma indicated that a profound inhibition (>80%) of the phosphorylation of ERK is required to obtain sufficient levels of clinical response with the *BRAF* inhibitor [139], and thus a modest inhibition of the RAF–MAPK pathway is related to the lack of efficacy in *BRAF*^V600E^-mutated mCRC treated with vemurafenib alone. Preclinical studies of mCRC revealed how *BRAF*^V600E^ tumors could alleviate the suppression of this signal transduction by the *BRAF* inhibitor, and suggested how this resistance could be overcome.

The MAPK signal transduction pathway, including RAS and BRAF, is normally regulated through several levels of feedback loops involving activated, phosphorylated ERK (pERK) [135,136,137,138,139,140,141,142]. Epidermal growth factor (EGF) binds to its specific receptor (EGFR) and initiates the signal transduction of this pathway. Activated RAS induces the dimerization of RAFs which are capable of phosphorylating downstream effectors, MEK and ERK [143,144,145,146]. The resulting pERK stimulates a subset of gene expression, but also inactivates EGFR and attenuates this signal transduction pathway [8,9] (Figure 1A). By contrast, in *BRAF*^V600E^-mutated tumors, the MEK–ERK axis is constitutively active, in a manner independent of the *RAS*-mediated dimerization of RAF [9,144], and is thereby refractory to the negative feedback circuit in this type of tumor (Figure 1B).

The use of an ATP-competitive *BRAF* kinase inhibitor, such as vemurafenib, revealed a transient suppression of the signal transduction and p-ERK levels [8,9,137]; however, the p-ERK level was recovered in the *BRAF* mutant cell line within 24 h [147]. The suppression of pERK reduces the power of the negative feedback circuit and preserves the activity of EGFR. This paradoxically activates RAS and the MEK–MAPK signaling axis, through the direct activation of another class of RAF, CRAF, or by facilitating BRAF–CRAF heterodimer formation [148,149,150,151,152,153,154,155] (Figure 1C). Given these alterations, a combinatory intervention of BRAF and EGFR and/or MEK seems promising [142,152] (Figure 1D). Corcoran et al. assessed the combined inhibition with dabrafenib (BRAF) and trametinib (MEK) in the treatment of *BRAF*-mutated mCRC. Among 43 patients, five patients achieved a partial or complete response and the median PFS was 3.5 months (95% CI 3.4–4.0 months) [155]. Compared to the BRAF inhibitor alone or standard chemotherapy, the dual MAPK signal pathway blockade using the BRAF and MEK inhibitors demonstrated modest efficacy [83,136,155]. The insufficient suppression of MAPK signaling was thought to be attributable to the limited response to the BRAF inhibitor [155]. Yaeger et al. conducted a pilot study to elucidate another dual blockade of the RAF–MEK–MAPK signaling pathway with Pmab and vemurafenib in the treatment of *BRAF*^V600E^-mutated mCRC [156]. For 12 patients treated with target lesions, only a subset of patients (*n* = 2) achieved a partial response (RR: 13%). The median PFS and OS were 3.2 months (95% CI 1.6–5.3 months) and 7.6 months (95% CI 2.1—not achieved months), respectively, and were still unsatisfactory [156].

Subsequently, another BRAF inhibitor, encorafenib, was developed and demonstrated encouraging results when combined with Cmab in BRAF-mutated mCRC patients [157]. The ORR was 19.2% and the median duration of response was 46 weeks in the doublet regimen (encorafenib and Cmab) group [157]. The first *BRAF*^V600E^ targeting therapy was established by the positive results of the BEACON CRC trial [4,5]. The BEACON CRC trial was conducted for previously treated cases of *BRAF*^V600E^-mutated mCRC. Patients were assigned to three groups, and the effectiveness and safety of the triplet-therapy group (encorafenib, binimetinib, and Cmab), and the doublet-therapy group (encorafenib and Cmab) were compared with the control group (Cmab + FOLFIRI or irinotecan). The primary endpoints were the OS and ORR of the triplet-therapy group, and secondary endpoints these of the doublet-therapy group [5]. The median OS was 9.0 months in the triplet-therapy group vs. 5.4 months in the control group (HR 0.52, 95% CI 0.39–0.70). The median OS in the doublet-therapy group was calculated to be 8.4 months (HR was 0.60, 95% CI 0.45–0.79) [5]. A significant improvement in ORR was also observed in the triplet-therapy group (26%, 95% CI 18–35% vs. 2%, 95% CI 0–7%; *p* < 0.001) [5]. Besides, an exploration of up-front use of the triplet therapy, titled ANCHOR CRC, is currently ongoing [158].

Thus, the *BRAF* mutation has been changed from a merely prognostic marker to a targetable gene mutation over the past two decades. Triplet therapy can be the first specific treatment for *BRAF*^V600E^-mutated mCRC, but there are several issues with this novel strategy that need to be resolved. First, whether MEK inhibitor is the optimal partner for BRAF inhibitor and anti-EGFR antibody should be addressed, because the PI3K/AKT signaling pathway exists downstream of receptor tyrosine kinases (RTKs), as well as the RAS–RAF–MEK–MAPK pathway, and there are interactions between these two downstream axes. The activation of the PI3K/AKT pathway is known to be involved in the mechanism of resistance to the BRAF inhibitor [151,155] (Figure 1E). A colorectal cancer cell line with the *PIK3CA* mutation or *PTEN* loss is more resistant to the BRAF inhibitor than wild-type cells. Mao et al. reported that the combination of a BRAF and PI3K inhibitor revealed synergistic anti-tumor activity in vitro and vivo [159,160]. A phase Ⅰ study of encorafenib and Cmab with alpelisib (a PI3Kα inhibitor) showed promising clinical outcomes (ORR: 17.9%, median PFS; 4.2 months, 95% CI; 4.1–5.4 months) [157]. Another observation questioning the combination is that triplet therapy using vemurafenib, Cmab, and irinotecan revealed favorable results in an early-phase clinical trial [161]. For 17 patients with *BRAF*^V600E^-mutated mCRC treated by triplet therapy, 14 patients (82.3%) exhibited tumor shrinkage and the ORR reached 35% (95% CI; 14–62%) [161]. This result indicated that irinotecan could be a candidate for treatment in combination with a BRAF inhibitor and anti-EGFR therapy. Moreover, a direct inhibition of ERK would potently suppress MAPK reactivation, regardless of the upstream signal [9,162]. Recently, Hazar-Rethinam et al. reported attractive results in vitro, demonstrating that convergent therapy using the ERK inhibitor could simultaneously overcome multiple heterogeneous resistant mechanisms. They proposed a novel up-front concept which can in theory eradicate the minority of resistant subclones that preexist in the population. Notably, their real-time cfDNA monitoring experiment demonstrated that triplet therapy targeting BRAF, EGFR, and ERK could potently suppress the outgrowth of subclones harboring RAS or MEK mutations, more strongly than the combination of BRAF, EGFR, and MEK inactivation [163]. This novel combination could lead to improved outcomes compared to those of the BEACON regimen; however, the treatment might also affect cells without the *BRAF* mutation. An anti-programed cell death 1 (PD-1) antibody could serve as a partner agent with a BRAF inhibitor and anti EGFR mAb. Recently, Corcoran et al. demonstrated the early results of their clinical trials (NCT03668431) using dabrafenib, trametinib and spartalizumab (anti-PD-1 antibody) in *BRAF*^V600E^-mutated mCRC patients at the ESMO World Congress on GI Cancers held in July 2020 [158]. They revealed promising data, i.e., ORR of 33%, in the patients with MSS (*n* = 17) and MSI-H (*n* = 8), which was superior to the previous result [158] (ORR 12%, dabrafenib and trametinib). Interestingly, this combination therapy was also effective for MSS CRC patients. This will raise the question of whether the anti-EGFR mAb or anti-PD-1/PD-L1 antibody is the preferred partner for the BRAF and MEK inhibitors. Further investigations for optimal combinations await.

Second, we should also pay attention to the toxicities by adding an MEK inhibitor and ask whether the clinical benefit still outweighs any negative side effects. The BEACON trial, while revealing the superiority of triplet therapy compared to the control (Cmab + FOLFIRI or irinotecan), could not evaluate the benefit of the triplet regimen over the doublet regimen for statistical reasons [5]. The following analysis of the BEACON trial reported the superiority of the triplet regimen in the ORR (27% vs. 20%, which was consistent with the previous report, but the median survival time was identical (9.3 months) between the triplet and doublet regimen [5,164]. Whereas adverse events, such as headache, musculoskeletal pain, arthralgia, and myalgia occurred frequently in the doublet group, gastrointestinal toxicities, such as diarrhea, nausea, and vomiting, were more frequently seen in the patients treated with the triplet therapy [5,164]. Although differences in the toxicity profile did not affect the treatment continuation or intensity [5], cases with severe headaches, musculoskeletal pain, arthralgia, and myalgia (grade ≥ 3) were rare (>1%) and severe-grade diarrhea was observed in 10% of patients with triplet therapy. Thus, the doublet therapy appears to be a less toxic regimen than triplet therapy. Corcoran et al. reported that the triplet therapy (dabrafenib (BRAF) + Pmab + trametinib (MEK)) showed a higher ORR (21% vs. 10% for doublet) and better tumor shrinkage than the doublet therapy (dabrafenib + Pmab) [165]. However, in terms of the disease control rate (DCR) and PFS, numerically similar results were observed for triplet (DCR: 86%, median PFS: 4.2 months) and doublet therapy (DCR: 90%, median PFS: 3.5 months) [165]. Considering these results, triplet therapy is not necessarily accepted as a standard regimen worldwide. In fact, according to the up-date results of the BEACON CRC study, the numerical difference of the ORR between the triplet (27%) and doublet (20%) were preserved, but the difference in OS was not observed. The principle investigator of this trial declared that only the doublet therapy for *BRAF*-mutated mCRC was considered to proceed for FDA approval [164], and triplet therapy would not be an option for all *BRAF*^V600E^-mutated mCRCs. Finally, in April 2020, the FDA approved encorafenib in combination with cetuximab for *BRAF^V600E^*-mutated mCRCs [166]. A careful assessment of patients in selecting triplet or doublet therapy is needed.

Third, the efficacy of triplet therapy might be modified by the MSI status. Among the patients treated with triplet therapy, the response rate of patients with MSI-H/dMMR tumors was 46% (95% CI 17–77%) and higher than that of MSS/pMMR (27%, 95% CI 17–77%). The PFS of patients with MSI-H/dMMR tumors exhibited a trend toward longer survival compared with that of MSS/pMMR (HR 2.62, 95% CI 1.00–6.91, *p* = 0.045) [165]. In addition, the MSI-H/dMMR tumor itself had already been established as a predictive biomarker in patient selection for anti-PD1 therapy [167,168,169]. Due to the very limited number of cases of *BRAF*-mutated MSI-H mCRC patients, it is presently too early to conclude whether the effectiveness of *BRAF* targeting therapy is affected by the *BRAF* status.

How can we possibly improve upon the vertical blockade strategy? Initially, insufficient suppression of the RAS–RAF–MEK–MAPK pathway, as described above, was thought to account for the unsatisfying results with the vertical blockade; however, pharmacodynamic analysis of paired tumor specimens, before and after treatment, revealed that the degree of MAPK pathway suppression is not correlated with the clinical outcomes [165]. On the one hand, doublet therapy with trametinib and Pmab (T+P) could reduce the pERK level significantly, but no responder to this doublet regimen was found. On the other hand, doublet therapy with dabrafenib and Pmab (D+P) could not reduce p-ERK sufficiently and the median p-ERK level of the D+P regimen was higher than that of T+P. Despite the weaker suppression of the MAPK signaling pathway, a higher response rate (10% vs. 0%) and longer PFS (3.5 months vs. 2.6 months) were observed in patients treated with the D+P regimen for unknown reasons [165]. Western blotting analysis of the phosphorylation of MEK or ERK may not be the right assay to evaluate the activity of this RAS–RAF–MEK–MAPK pathway. The plastic and reversible nature of this pathway makes it difficult to overcome the associated malignancy. Resistance can be caused by the compensatory reactivation of the MAPK pathway, through the acquisition or preexistence of RTK amplification (not only *EGFR*, but also *HER2*, *HER3* or *MET*), *RAS* mutation or *RAS* amplification, *BRAF* amplification, and *MEK* mutation, which could cause resistance to the BRAF inhibitor [8,9,151,163,165,170,171,172,173]. For instance, while the *BRAF* and *RAS* mutation are initially found in a mutually exclusive manner, these tumors additionally acquire *NRAS* or *KRAS* mutations after vemurafenib treatment, as indicated by the analysis of their circulating tumor DNA (ctDNA) [138]. In addition, Pietrantonio and Oddo et al. reported the case that combination therapy with crizotinib (MET inhibition) could transiently overcome the acquired resistance after BRAF inhibition, but MET hyper-amplification emerged as a second acquired resistance [172,173]. It seems therefore that the heterogeneous and plastic nature of the RTK signaling pathway prevents us from conquering this malignancy.

Paradoxically, however, these alterations underlying adaptive resistance can be toxic in the absence of treatment. A preclinical study of melanoma with *BRAF*-mutation indicated that cells with the acquired upregulation of EGFR expression were induced by using a BRAF inhibitor, but these cells became vulnerable when the treatment was ceased. The hyperactivated state of the RAS–RAF–MEK–MAPK pathway induced by the overexpression of EGFR led the cells into oncogene-induced senescence when the BRAF inhibitor was deprived [174]. The “addiction” to the BRAF inhibitor led to successful re-challenge cases of BRAF inhibition, i.e., the restored sensitivity, after a drug holiday [175]. Thus, using the optimal method and timing to evaluate the activity of this signaling pathway seems to be an important issue, and its “real-time monitoring” should be useful.

In summary, a novel triplet therapy with encorafenib, binimetinib, and Cmab emerged as the targeting therapy for *BRAF*^V600E^-mutated mCRC and has been used since 2020. Given the presence of specific treatment in clinical practice, *BRAF*^V600E^-mutated mCRC has become an independent subset of mCRCs. The constitutive activation of RAF–MEK–MAPK signaling is central to supporting the survival of *BRAF*-mutated cells in mCRC. Unlike melanoma, BRAF inhibition alone revealed limited efficacy, and acquired the alteration-mediated reactivation of the MAPK pathway which underlies the resistance mechanism to BRAF inhibitors. Therefore, comprehensive regulation of this pathway is required to conquer *BRAF*^V600E^-mutated mCRC. Bearing this in mind, a vertical blockade of signal transduction is a promising way to achieve a clinical response in mCRC but it still needs improvement. These resistance mechanisms seem to stem from the plastic and diverse nature of the RAS–RAF–MEK–MAPK pathway. Moreover, cancer cells opt to use signaling networks outside the RAF–MEK–MAPK pathway in an attempt to evade the vertical blockade strategy. The strategy to completely and effectively control *BRAF*-mutated mCRC is yet to be developed.

## 7. Future Perspectives on *BRAF*-Mutated mCRC and Beyond

Recently, experts in this field have paid attention to the heterogeneity of *BRAF*-mutated mCRC. All *BRAF*-mutated mCRCs do not necessarily belong to the same subset of mCRCs, and there are several variations in terms of DNA sequencing, RNA expression, kinase activity, clinicopathological features, sensitivity to anti-EGFR therapy, and prognosis. In this final chapter, we would like to describe the future perspectives, focusing on the accumulating data on the variations of *BRAF-*mutated mCRCs.

One of the explorations on the heterogeneity of *BRAF*-mutated CRCs was focused on the outside of codon 600 of *BRAF—*the *BRAF^non-^*^V600E^ mutation. More than 1000 unique *BRAF* mutations have been previously described in patients with various malignancies [176]. Recent advances in NGS technology have found several variants of the *BRAF^non-^*^V600E^ mutation, which account for approximately 2% of mCRCs. There are several differences in the clinicopathological features and prognosis between V600E and non-V600E mutations [177,178,179,180]. Patients with *BRAF^non-^*^V600E^ mutations were found to be younger, have left-sided primary disease, have a non-mucinous histology, and have less peritoneum metastasis compared to those with *BRAF*^V600E^ mutations. Moreover, different molecular features were also reported for *BRAF^non-^*^V600E^ and *BRAF*^V600E^ mutations. *BRAF^non-^*^V600E^ mutations had less association with MSI-H tumors, and some *BRAF^non-^*^V600E^-mutated CRCs even had a co-occurrence with *RAS* mutations [177,178,179,180]. Not all these rare variants of *BRAF* mutations had been proven to have oncogenic activities in vivo/vitro. Constitutive activation of the RAF–MEK–MAPK pathway is thought to be an essence of the malignant nature of *BRAF*^V600E^ mutations. However, kinase activity toward the phosphorylation of the MAPK pathway is impaired or even lost in some *BRAF^non-^*^V600E^-mutated CRCs [144,181]. Oncogenic RAS is required for kinase-dead BRAF to drive tumor progression [151]. Recently, the novel classification of *BRAF* mutations based on kinase activity was proposed, including class 1 (activating *RAS*-independent *BRAF* mutations signaling as monomers), class 2 (activating *RAS*-independent *BRAF* mutations signaling as dimers with CRAF), and class 3 (*RAS*-dependent *BRAF* mutations with impaired kinase activity or kinase-dead) [182,183,184]. This proposal was, to some extent, clinically accepted to reflect the sensitivity to anti-EGFR antibodies. Patients with class 3 *BRAF* mutations potentially respond to anti-EGFR therapy [182,184].

Gene expression is controlled by several factors in terms of genomic and epigenomic aspects. Classification by gene expression analyses was thought to be the consequence of these complexities. Therefore, categorization based on gene expression could have more links to tumor behaviors and identify more biologically homogeneous subgroups than DNA sequencing. Previously, several studies have reported that classifications based on gene expression analyses have associations with clinicopathological features, predictions for treatment, or prognosis in other cancers [185,186,187,188]. Recently, the Colorectal Cancer Subtyping Consortium (CRCSC) suggested four consensus molecular subtypes (CMSs) of CRCs based on international transcriptomic data at the largest scale [189]. In their subtyping, *BRAF*-mutated CRCs were more enriched (>70%) in CMS1, which is characterized by high immunogenicity with a better prognosis. However, more than half of CMS1 cases were composed of *BRAF* wild CRCs. More recently, Barres et al. segregated *BRAF^V600E^*-mutated CRCs into two subtypes, called *BRAF* mutation (BM) 1 and 2, based on a gene expression analysis using a dataset of 218 *BRAF^V600E^*-mutated CRC patients. The BM1 subtype is highly active in KRAS/mTOR/AKT/4EBP1 signaling and the BM2 subtype is associated with cell cycle check point-related genes [190]. In addition, a recent study showed that the BM1 subtype was more sensitive to the triplet therapy with dabrafenib, trametinib, and panitumumab, and had better clinical outcomes compared to the treatment of the BM2 subtype. The BM subtype was an independent predictive factor in the multivariate analysis that includes other immune-related signatures [191]. Whether the BM subtyping is a robust biomarker, it requires future validation to be determined. This result indicated the transcriptional context could affect the significance of gene mutation. A paradigm shift from “one-gene, one-drug” based on a single driver mutation to “multi-gene, multi-drug”, which is expected regarding the future perspective of precision medicine [192].

Other approaches using gene expression data have been conducted to define the *BRAF* mutation signature and identify the subgroup of CRCs with *BRAF*-like malignant potential, without, having the *BRAF* mutation itself, in melanoma and breast cancer ahead of CRC [192,193,194]. Tian et al. reported that *BRAF*-mutated CRCs displayed more distinctive expression patterns than those of *KRAS* and *PIK3CA* mutations [195]. Popovici et al. identified *BRAF*-mutant like CRCs in *KRAS* mutations or double wild-type patients, with a poor prognosis similar to that of *BRAF*-mutated CRC patients [196]. Furthermore, Vecchione et al. conducted a functional analysis of *BRAF*-mutant-like cell lines and discovered the common vulnerability that the loss of *RANBP2* was lethal to *BRAF*-mutant-like cells. In addition, they demonstrated the sensitivity of *BRAF*-mutant-like cells to vinorelbine in vitro and vivo. Surprisingly, they obtained an archival sample of the super responder which was enrolled in the clinical trial of vinorelbine for CRC in the 1990s. Notably, that tumor showed a *BRAF*-mutant-like expression pattern [197].

Moreover, the heterogeneity of *BRAF*-mutated CRCs has been reported in several studies by more practical and easily available methods, such as the immunohistochemistry of clinicopathological features. Several studies have reported the differences among *BRAF*-mutated CRCs in the methylation phenotype (CIMP-high and CIMP-low), expression of the cytokeratin (CK7 and CK20), the transcription factors (CDX2) and immunohistochemical (IHC)-based CMS classification [198,199,200,201,202]. Furthermore, Loupakis et al. suggested a more practical classification method by using data which were easily available in daily practice, such as performance status (PS) and laboratory data only. Interestingly, they revealed that the prognosis of *BRAF*-mutated CRCs was clearly divided according to their clinical subtyping [203]. These results indicated that *BRAF*-mutated CRCs were not one disease but that there was inter-tumor heterogeneity among *BRAF*-mutated CRCs.

In summary, several researchers have reported the heterogeneity of *BRAF*-mutated CRCs. Patients with *BRAF*^non-V600E^ mutations have different clinicopathological features compared to those with *BRAF*^V600E^ mutations. Moreover, they could benefit from anti-EGFR antibodies. Different kinase activities of V600E and non-V600E in the RAF–MEK–MAPK pathway could explain the sensitivity to anti-EGFR therapy. Classification based on gene expression analysis could broaden the horizon of therapeutic targets for BRAF-mutant-like CRCs without any *BRAF* mutations. Currently, the results of these phenotyping studies are not consistent.

## 8. Summary and Conclusions

Since the first report of the *BRAF* mutation in human cancer was published in 2002, nearly two decades ago, the significance of *BRAF*-mutated CRC has been established in clinical practice, changing from being merely a prognostic biomarker marker, via being considered a moderate predictive marker for anti-EGFR therapy and actionable genetic alteration, and toward becoming a targetable mutation. As described in this review, cancer precision medicine based on driver mutations has accelerated the emergence of *BRAF*-mutated CRCs from basic research to clinical practice. Several research studies on the resistance mechanism of *BRAF* targeting therapy have demonstrated the importance of a profound understanding of the RAS–RAF–MEK–MAPK pathway in conquering this malignant tumor. Therefore, especially in CRC, we should recognize the therapeutic target as being not only the *BRAF* mutation alone, but also the entire RAS–RAF–MEK–MAPK signaling pathway. Combination therapy is necessary to overcome the resistance mechanism associated with multiple genes (Figure 2).

The *BRAF* mutation is also a tumor agnostic genetic alteration and therefore it could become targetable, regardless of the tumor origin. Indeed, current precision medicine based on the actionable genetic alteration has precisely attacked the target, but it cannot always predict the tumor’s Achilles heel. Driver mutation-guided clinical trials could provide additional therapeutic options for selecting patients, but these efficacies are far from being satisfactory. Inter- and intra-tumor heterogeneity always arises as a hazard in the treatment of *BRAF*-mutated CRCs. Several findings have indicated the variety and plasticity of this type of tumor which underly the resistant mechanisms. The variety and plasticity are derived from the mechanism acquired by normal cells to adapt to their changing environments for surviving at their original site. Therefore, understanding their tissue-specific natures is necessary for defeating this malignancy. Further investigations into the nature of the origin, including the surrounding immune cells, represent the future direction for the next developments in optimal precision medicine.

## Figures and Tables

**Figure 1 cancers-12-03236-f001:**
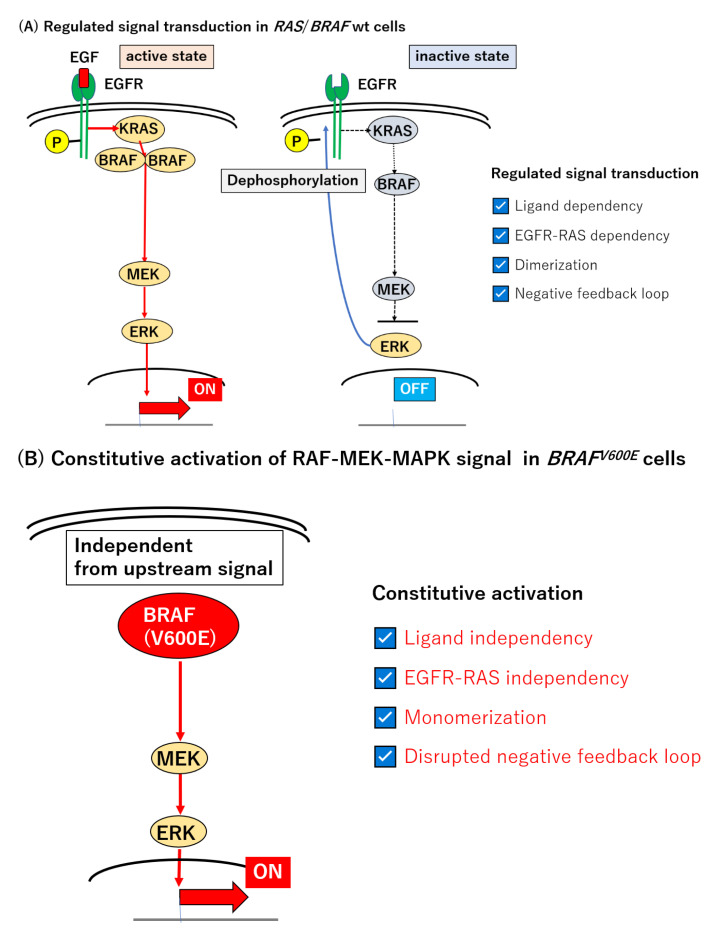
Signal transduction in the EGF/EGFR–RAS–RAF–MEK–MAPK pathway. (**A**) Regulated signal transduction in *RAS*/*BRAF* wt cells. EGFR is activated by the binding of EGF, a specific ligand to become an active homodimer. Homodimerization stimulates the autophosphorylation of tyrosine residues in the EGFR domain, which initiates the downstream signal cascade via RAS–RAF–MEK–MAPK signal transduction. Activated RAS-induced dimerization of RAFs is necessary for phosphorylating downstream effectors, MEK and ERK (left). This signal transduction pathway comprises a negative feedback loop, i.e., elevated pERK in turn causes the de-phosphorylation of EGFR and thereby attenuates this signal (right). (**B**) The constitutive activation of the RAF–MEK–MAPK signal in *BRAF^V600E^* cells. *BRAF^V600E^*-mutated cells constitutively activate its downstream effectors as a monomer in the absence of upstream signals. Given this independence, the activation of the MAPK pathway is refractory to the pERK-mediated negative feedback regulation. This explains why RAF–MEK–MAPK signal transduction is constitutively active in BRAFV600E cells. (**C**) BRAF inhibition with ATP competitive RAF inhibitors in *BRAF^V600E^*. ATP-competitive inhibitors such as vemurafenib act on monomeric BRAF and suppress downstream signal transduction. These kind of kinase inhibitors can initially control the constitutive activation; however, because of the reduction of pERK, they attenuate the negative feedback. EGFR-mediated signal transduction will be reactivated before long in CRCs, as activated RAS induces RAF dimerization, between BRAF and CRAF, which is refractory to vemurafenib. Therefore, *BRAF*-mutated CRCs can easily overcome the monomer inhibition treatment. (**D**) The concept of the vertical blockade strategy: blocking both upstream and downstream of BRAF using anti-EGFR mAb and MEK inhibitor in EGFR signal axis, respectively, were developed to overcome the resistance mechanism. Theoretically, this vertical blockade can suppress the growth signaling if and only if the EGFR–RAS–MEK–MAPK pathway remains the prime pathway throughout treatment. (**E**) A possible signaling pathway providing resistance to the vertical blockade triplet therapy. The EGFR signal axis is not the only pathway for the proliferation of BRAF-mutated CRCs. Reactivation of other receptor tyrosine kinases (RTKs), such as HER2 and MET, was pointed out in vitro. Moreover, the RAS–RAF–MEK–MAPK pathway is known to interact with the PI3K/AKT pathway. *BRAF*-mutated CRCs can therefore evade the vertical blockade with triplet therapy through these alternative means. EGFR: epidermal growth factor receptor, EGF: epidermal growth Factor, Wt: wild type, pERK: phosphorylated ERK, mAb: monoclonal antibody, CRC: colorectal cancer.

**Figure 2 cancers-12-03236-f002:**
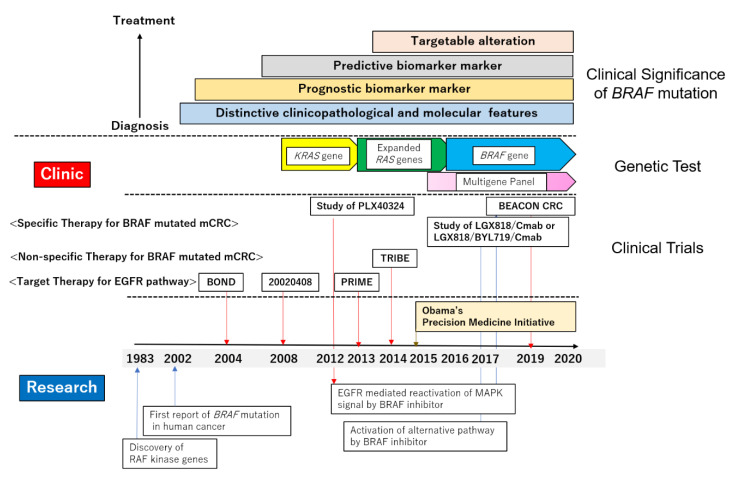
History of basic studies, genetic tests, and clinical trials that together established the significance of the BRAF mutation in the treatment of CRCs. The *RAF* genes, *ARAF*, *BRAF,* and *CRAF,* were discovered in 1983. The first report of *BRAF* mutation in human cancers was published in 2002 and included melanoma, lung cancer, and colorectal cancer. Early studies on *BRAF* mutation in CRC were focused on its characteristic molecular and clinicopathological features. Although *BRAF* mutation had a strong negative impact on the survival, the presence of the *BRAF* mutation itself had little influence when making decisions on treatment options at this time. At that stage, the *BRAF* genetic test was not mandatory in clinical practice. Since the success of clinical trials with anti-EGFR mAbs in 2004, targeting the EGF/EGFR pathway became technically feasible in treating metastatic colorectal cancer (mCRC). Unlike for the *KRAS* mutation, the *BRAF* mutation alone was insufficient to determine indication or anti-EGFR mAb therapy. Studies using clinical samples pointed out a loss of EGFR-mediated negative feedback to be the mechanism of resistance to the BRAF inhibitor. Combination therapies suppress not only activated BRAF but the entire RAF–MEK–MAPK signal transduction pathway, successfully controlling the *BRAF*-mutated mCRC. ARAF, BRAF, CRAF: Raf murine sarcoma viral oncogene homolog A, B, C; EGFR: epidermal growth factor receptor; mAb: monoclonal antibody; CRC: colorectal cancer.

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
