# Peer review of "BRAF Mutation in Colorectal Cancers: From Prognostic Marker to Targetable Mutation"

_cancers, 2020, doi:10.3390/cancers12113236_

Round 1

Reviewer 1 Report

Further to my previous comment 1, actually two studies reported the association between ethnicity and Braf mutation rate Harry H Yoon et al 2015 and Dallas R. English et al 2018. I think the authors can include their findings in paragraph 3:Distinctive characteristics of BRAF-mutated CRC: molecular and clinicopathological aspects.

Author Response

This manuscript is a resubmission of an earlier submission. The following is a list of the peer review reports and author responses from that submission.

Round 1

Reviewer 1 Report

This is an interesting and timely review, nicely summarizing all we know about BRAF in mCRC. Although the topic is ocmplex and complicated, the authors have managed to provide a logical structure to the article.

However, many expressions do not conform to standard (medical) English and extensive review by a native English speaker is strongly recommended.  

Reviewer 2 Report

This article provides a comprehensive review on the clinical significance of BRAF mutation (mainly V600E) in colorectal cancer patients. Their review summarized the clinical trial findings of different treatment regimen and provided suggestions on perspective study. These are informative for clinicians for designing the treatment for CRC patients.

I only have two minor comments:

1. Is there any correlation between BRAF mutation and ethnicity, and hence difference on prognosis?

2. Can the author discuss further on the potential treatment efficacy of BRAF mutant-based inhibitor plus immunotherapy (such as PD-L1) on MSS CRC patients?

Reviewer 3 Report

I commend the authors for a thorough review on the clinical significance of BRAFV600E mutation as prognostic and predictive biomarker, as well as a therapeutic target in colorectal cancer.

Comments:

  1. The title should be changed to “BRAF mutation in colorectal cancer………” as the authors also included a review on the prognostic impact of BRAF mutation in non-metastatic (curatively resected) CRC
  2. I find parts of the manuscript difficult to follow and some sentences challenging to understand. For example:- section 4 (BRAF mutation as a negative prognostic marker) line 100 “However, we will discuss the prognostic impact of the BRAF mutation in resectable stage CRC in this review.” The authors then went on to discuss the prognostic significance of BRAF mutation in metastatic disease from line 178. Also line 336 – “Clinical sequencing for searching for the driver genetic alterations has been widely conducted. Molecular targeting agents, mainly inhibitors, were used as a bullet for beating the driver and, umbrella- or basket-type clinical trials became a saucer for the results of these genetic tests.” I find these sentences difficult to understand.
  1. Line 249 – “Clinicians should consider the BRAF status as one of the other clinicopathological factors. Clinicians recognized the significance of the BRAF mutation, but accumulating data could not show a robust reason to distinguish BRAF mutated CRCs in clinical practice.” I disagree with these statements. As the authors themselves mentioned later on in the manuscript, given BRAF mutation is associated with poor prognosis and shorter 2nd line PFS, it is important to identify patients with this mutation at the time of metastatic disease diagnosis as this would guide the use of intensive triplet FOLFOXIRI treatment upfront, and also plan for BRAF targeted therapy in the 2nd line.
  2. Line 343 – “In summary, the BRAF mutation was shown to have a potential negative 343 predictive value for treatment with anti-EGFR antibody, but evidence that emerged from clinical trials was insufficient for obtaining the consensus that upfront BRAF gene testing should be mandatorily performed before anti-EGFR therapy like KRAS and expanded RAS”. I disagree with this comment – as mentioned previously, identifying BRAF mutant mCRC upfront is useful to guide the first-line chemotherapy regimen, i.e. doublet vs triplet. Additionally, given most hotspot mutation sequencing is now done as a panel, it would be more efficient to perform both the RAS and BRAF mutation testing together. NCCN guideline is now recommending testing BRAF mutation at diagnosis of mCRC.

5. line 632 – “Moreover, the heterogeneity of BRAF-mutated CRCs has been reported in several studies by more practical and easily available methods, such as the immunohistochemistry of clinicopathological features”. – the authors should expand on this statement.